# Targeting Tyrosine Kinases in Ovarian Cancer: Small Molecule Inhibitor and Monoclonal Antibody, Where Are We Now?

**DOI:** 10.3390/biomedicines10092113

**Published:** 2022-08-29

**Authors:** Aimee Rendell, Isobel Thomas-Bland, Lee McCuish, Christopher Taylor, Mudra Binju, Yu Yu

**Affiliations:** 1Curtin Medical School, Faculty of Health Sciences, Curtin University, Perth, WA 6102, Australia; 2Faculty of Health Sciences, Curtin Health Innovation Research Institute (CHIRI), Curtin University, Perth, WA 6102, Australia; 3Division of Obstetrics & Gynaecology, University of Western Australia Medical School, Perth, WA 6009, Australia

**Keywords:** ovarian cancer, tyrosine kinases, platinum resistance, recurrent cancers, cancer vaccine, kinase inhibitors

## Abstract

Ovarian cancer is one of the most lethal gynaecological malignancies worldwide. Despite high success rates following first time treatment, this heterogenous disease is prone to recurrence. Oncogenic activity of receptor tyrosine kinases is believed to drive the progression of ovarian cancer. Here we provide an update on the progress of the therapeutic targeting of receptor tyrosine kinases in ovarian cancer. Broadly, drug classes that inhibit tyrosine kinase/pathways can be classified as small molecule inhibitors, monoclonal antibodies, or immunotherapeutic vaccines. Small molecule inhibitors tested in clinical trials thus far include sorafenib, sunitinib, pazopanib, tivantinib, and erlotinib. Monoclonal antibodies include bevacizumab, cetuximab, pertuzumab, trastuzumab, and seribantumab. While numerous trials have been carried out, the results of monotherapeutic agents have not been satisfactory. For combination with chemotherapy, the monoclonal antibodies appear more effective, though the efficacy is limited by low frequency of target alteration and a lack of useful predictive markers for treatment stratification. There remain critical gaps for the treatment of platinum-resistant ovarian cancers; however, platinum-sensitive tumours may benefit from the combination of tyrosine kinase targeting drugs and PARP inhibitors. Immunotherapeutics such as a peptide B-cell epitope vaccine and plasmid-based DNA vaccine have shown some efficacy both as monotherapeutic agents and in combination therapy, but require further development to validate current findings. In conclusion, the tyrosine kinases remain attractive targets for treating ovarian cancers. Future development will need to consider effective drug combination, frequency of target, and developing predictive biomarker.

## 1. Ovarian Cancer: Current Understanding and Treatment

Ovarian cancer (OC) is a leading cause of cancer-related death in women with the second highest mortality rate of gynaecological cancers worldwide [1]. In 2020, an estimated 313,959 cases were diagnosed worldwide, accounting for 3.4% of all new cases of cancer in women and an age-standardised incidence rate of 6.6 per 100,000 [1]. In the same year 207,252 deaths were recorded, accounting for 4.7% of women cancer related deaths with an age-standardised rate of 4.2 per 100,000 [1].

Diagnosis of OC follows the guidelines suggested by the International Federation of Gynaecology and Obstetrics (FIGO), consisting of a four-level staging system based on tumour location, histological profile, and level of metastatic activity [2]. Because early-stage diseases are often asymptomatic, around 75% of cases are only diagnosed at advanced stages III and IV, where tumours have disseminated and invaded the abdominal cavity beyond the pelvic region [2,3]. The highly invasive nature of OC results in the progression of the disease beyond the ovary, to the surrounding peritoneum, regional lymph nodes, and other organs within the peritoneal cavity. Even at an advanced stage of cancer, most common symptoms are nonspecific and include abdominal discomfort and bloating [3,4].

An estimated 85–90% of OC are of epithelial origin [5]. These are multicentric in nature and potentially include mesothelial invaginations, endosalpingiosis, and pelvic peritoneum. OC is highly heterogeneous and can be classed into subtypes based on histological profiles [2]. The most common form of OC is high grade serous ovarian carcinoma (HGSC), which is frequently studied in clinical trials. Other known histological subtypes of OC include clear cell, endometrioid, and mucinous carcinomas, as well as low-grade serous ovarian cancer (LGSC) [2,6]. Zhou et al., have suggested that histological subtypes are correlated with survival outcomes in patients. They found that 5-year overall survival was lower in patients with mucinous and clear cell subtypes (14.2% and 18.8%, respectively) as compared to HGSC and endometrioid (28.1% and 38.6%, respectively) [7].

Approximately half of the tumours also carry gene mutations causing defects in homologous recombination [8]. The mutational profiles of every histological subtype are different: HGSC usually carries TP53 mutations, LGSC and mucinous have increased in BRAF and RAS mutations, while endometrioid and ovarian clear cell carcinoma (OCCC) have mutations in ARID1A and PIK3CA [9,10]. Furthermore, transcriptomic analyses have demonstrated additional molecular subtypes of HGSC into mesenchymal, immunoreactive, differentiated, and proliferative [11,12], although the robustness and clinical utility of molecular subtyping remains obscure [11,13,14]. 

The standard first line treatment for OC involves the combination of surgical intervention and chemotherapy-based treatment [15]. Surgical intervention involves the practice of debulking, removing macroscopic tumoral deposits where possible to completely resect visible disease. Although the accuracy of tumour removal is often a strong predictor for overall survival (OS) [16], 50–60% of patients with advanced OC will require further treatment [17]. The common therapeutics administered are carboplatin and paclitaxel. Platinum/taxane chemotherapy is administered intravenously following debulking surgery over six cycles every 21 days [15]. If deemed fit, a patient may also be exposed to intraperitoneal chemotherapy [18].The late presentation and diagnosis, as well as the heterogeneous nature of OC results in the overall 5-year survival rate of 30–45% [19,20,21].

Despite the high success of first line treatment, 70% of advanced stage patients experience a relapse [22]. The classification of recurrence is based on the platinum-free interval (PFI), ranging from highly sensitive to platinum refractory, based on time taken for new tumours to occur post chemotherapy [15]. Platinum sensitive patients (>6-month period) can undergo further rounds of chemotherapy treatment. For platinum resistant and refractory patients, there is currently no evidence-based standard for a second line of treatment. These patients can be considered for experimental therapy in clinical trials settings [23]. Combination of chemotherapy and other cytotoxic agents has been clinically used to prolong this progression-free period [15]. Whilst the exact mechanism for the progression of OC during treatment has yet to be established, it is widely accepted that the cell signalling pathways of receptor tyrosine kinases may impact and influence the dynamic nature of these malignant cells [24]. With limited effective treatment available for chemoresistant or refractory patients, there is a critical need for the establishment of therapeutic agents. Small molecule inhibitors such as tyrosine kinase inhibitors (TKIs) and monoclonal antibodies (mAb) targeting tyrosine kinase have been implemented in the maintenance of tumour sensitivity and in cytoreduction in a variety of cancers including breast cancer [25], gastrointestinal stromal tumours, and renal cell carcinomas [26]. Several therapeutics of this nature have been trialled in preclinical and clinical settings to determine their efficacy and adverse effects in the treatment of OC in both single agent and combination usage. 

## 2. Receptor Tyrosine Kinases

Receptor tyrosine kinases (RTKs) are enzymes tasked with the mediation of intercellular communication throughout the body [27]. RTKs exhibit their mode of action through the transfer of a phosphate group from ATP to one or more tyrosine residues, resulting in the conformational change of the target proteins [28,29]. Although tightly regulated under normal homeostatic conditions, RTK activity in cancer is highly dynamic, with altered functional roles due to mutations and overexpression [28]. The change in RTK function has been linked to malignancy in numerous cancers, including OC. The oncogenic activity of RTKs is assumed to be essential in the proliferation and progression of OC tumours [30,31]. RTKs rarely act in isolation, forming intricate co-activation networks with other TKs, diversifying signalling outcomes and preparing for ready adaptation should a pathway face interference [32]. This intercellular crosstalk is modified in cancers, with the plasticity of signalling demonstrated with chemoresistance and disease progression under therapeutic conditions [32,33]. Oncogenic activation of RTKs and their intermediates typically occurs through one of the following mechanisms: autocrine activation of RTK signalling (Figure 1a), chromosomal rearrangements (Figure 1b), duplication of tyrosine kinase domain (Figure 1c), gain of function mutations (Figure 1d), or genomic amplifications (Figure 1e) [24,27].

## 3. Epidermal Growth Factor Receptors

A common family of RTKs often observed as overexpressed in ovarian cancer is the epidermal growth factor receptor (EGFR) or ErbB family. Consisting of EGFR (also referred to as HER1/ErbB1), HER2 (ErbB2), HER3 (ErbB3), and HER4 (ErbB4), this family of structurally related receptors are essential in normal cellular development, proliferation, differentiation, and survival [34]. When the EGFR is abnormally activated, the dimerization or over-expression of ligand-dependent receptors leads to the manifestation of tumours which are epithelial in origin [35]. The ErbB receptor signalling occurs via MAPK, STAT, PI3K, and mTOR pathways, important for cell survival, proliferation, and differentiation, assisting in the oncogenic activity of the ErbB family [36]. The number of OC with EGFR activating mutations and amplification are small (<4% and 4–22%, respectively) compared to other cancers such as lung cancers [37]. EGFR overexpression, on the other hand, varies from 9–62%, depending on assay conditions and selected cut-off values [37]. While EGFR is commonly referenced in the literature, there is limited knowledge regarding associations between EGFR expression and disease outcome [38].

The association between HER2 expression and OC prognosis has been investigated, with HER2 amplification being non-prognostic [39]. A meta-analysis of 34 studies involving 5180 OC patients showed that expression of HER2 was negatively correlated with overall survival (OS) [40]. More recent data studying HER2 and HER3 in the same cohort of 105 cases suggest 3.8% positivity of HER2 by immunohistochemistry and 5.7% by in situ hybridization. In contrast, HER3 levels were higher, being 12.4% and 8.6%, respectively. This study did not identify significant correlation between HER2 status and survival. However, the HER3 status by fluorescence in situ hybridization was associated with poor progression-free survival (PFS) [41].

## 4. Vascular Endothelial Growth Factor Receptor and Its Pathways

A study has used 339 primary ovarian tumours to show that vascular endothelial growth factor (VEGF) was overexpressed in only 7% of tumours and was correlated with significantly poorer survival [42]. This suggests that the therapeutic success of targeting the VEGF pathway in isolation is limited to only a small subset of patients. 

Subsequent studies that examined VEGF receptors and their ligands compared primary high grade serous ovarian tumours and their paired distant omental metastases. The study showed that the protein expressions of VEGF-A, VEGF-D, and VEGFR1 were higher in the metastases than the primary lesions [43]. Significant indicators associated with short PFS include high VEGF-C level, low VEGFR3, and low epithelial expression of VEGF-A. The reasoning for these associations were not clear, although the survival analysis was univariate.

## 5. Platelet-Derived Growth Factor Receptors

Platelet-derived growth factor (PDGF) and PDGFR-α expression were mostly detected in the malignant ovarian tumours as compared to the benign tumours and the normal ovaries through immunohistochemistry. PDGFR-ß was not detected in either the normal epithelium or the tumour cells. Patients with PDGFR-α positive tumour cells were associated with shorter survival compared to negative tumours, suggesting the prognostic role of PGDFR-α in OC [44]. More recent investigation suggested that PDGFR-α and PDGFR-ß were not related with FIGO stage, grade, or histopathological subtype of OCs; but PDGFR-ß expression in cancer cells was associated with improved OS in 52 cases. The significance of this correlation in relation to therapy design is still yet to be fully understood [45].

## 6. c-MET

A meta-analysis study on c-MET in OCs involving 568 patients from seven studies suggested that patients with high levels of c-MET in tumours were associated with worse survival than those with low tumour c-MET level [46]. Although not statistically significant, there was a trend of c-MET overexpression associated with higher FIGO stage and lymph node metastases. Hence, c-MET expression may serve as a prognostic marker and may be useful to target in OC [46]. Other studies concluded that c-MET expression was greater in OCCC than the serous carcinoma subtype [47].

## 7. Other Emerging RTKs

A key RTK family influencing the cellular behaviour of OC is the Axl family; however, the inhibition of Axl as treatment has mainly been explored preclinically [48]. This family consists of three RTKs, namely Tyro3, Axl, and MerTK (or TAM receptors) [49]. Within the literature there are several alternate names for each receptor. Tyro3 is also referred to as c-Eyk, c-mer, MER, RP38, and Tryo12. Axl can also be called ARK, JTK11, Tyro7, and UFO. Tyro3 is also known as BYK, Etk-2, DTK, Rek, RSE, Sky, and Tif. Axl overexpression has been detected in multiple cancers, known to have a supportive role in tumorigenesis [50]. In chemoresistant OC cell samples, Axl has been observed to be expressed in high levels [48]. Ligand activated Axl promotes proliferation in tumours and its overexpression is correlated with poor prognosis in patients with OC [48]. Various studies have demonstrated the correlation between chemoresistance in OC cells whereby the inhibition of Axl resulted in an increased chemosensitivity of the HGSC cells to platinum-based therapeutics [48,51]. In the same RTK family, Tyro3 was found to be upregulated in Taxol-resistant cell lines, with a reported role of promoting proliferation in OC cells [52]. The inhibition of Tyro3 produced a reversal in the chemoresistance cell lines, restoring sensitivity to Taxol [49,53]

Another RTK target with promising preclinical results and potential influence over chemoresistance in OC is the receptor tyrosine kinase-like orphan receptor (ROR) family. Recent studies have demonstrated trends of elevated ROR1 and ROR2 in chemoresistant HGSC cell lines [54,55]. Forming part of the Wnt signalling pathway, the receptors have an important role in epithelial-mesenchymal transition, alluding to potential metastatic and chemoresistant influences [56]. Studies conducted by Henry et al., further established a strong correlation between these orphan receptors, with combined inhibition of both ROR1 and ROR2 demonstrating a significant chemo-sensitising effect to cisplatin in OC cell lines [54,55]. Whilst further study will be required to dissect the exact mechanism of each receptor in isolation, the ROR family represents a promising new target for potential therapeutic interventions.

## 8. Small Molecule Tyrosine Kinase Inhibitors

Tyrosine kinase inhibitors (TKIs) are small molecule pharmacologic agents which disrupt the signalling pathways of protein kinases through varying modes of inhibition [57]. As a result, TKIs are classified as ATP-competitive, also known as Type I, and non-ATP competitive, also known as Type II and Type III [57,58]. Nearly all TKIs are taken orally, with drug-specific therapeutic loading and dosage intervals. Common adverse outcomes of TKIs include nausea, fluctuations in blood pressure, skin disorders, fatigue, and diarrhoea, although symptom occurrence and severity vary greatly between different drugs and dosages [59]. Here, we describe TKI implication in OC treatment and maintenance. The clinical trials results are summarised in Table 1.

### 8.1. Small Molecule ATP-Competitive Tyrosine Kinase Inhibitors

ATP-competitive TKIs, also known as Type I TKIs, emit their mode of action through competing with ATP for phosphorylation binding sites. The majority of current TKIs in use are classified as Type I [57]. Effective Type I TKIs require high specificity to the target kinase in order to compete at the ATP-binding sites. The high amount of variation between kinase families in combination with the high rate of competition with ATP poses difficulties in the development of effective Type I TKIs [57,79].

#### 8.1.1. Sorafenib

Sorafenib has previously been shown to have anti-proliferative effects in thyroid cancer [80], hepatocellular [81], and renal cell carcinomas [82], via its inhibitory effect on VEGFR, PDGFR, Flt, c-KIT, b-raf, and c-raf [60]. As a maintenance therapy, sorafenib has shown limited efficacy in improving the PFS and OS in patients with OC. A phase II study evaluated the tolerability and efficacy of daily 400 mg of oral sorafenib in 59 patients with persistent or recurrent epithelial ovarian cancer (EOC) [60]. Median PFS and OS were 2.1 months (95% CI: 1.87–3.42 months) and 16.3 months (95% CI: 11.1–22.2 months), respectively. Furthermore, 33.9% (20 of 59) of patient had stable disease (SD) at best response and 3.4% (2 of 59) had a partial response (PR); moreover, 23.7% of patients remained free of disease progression for at least 6 months. However, there was limited efficacy of sorafenib in patients with ascites and it was associated with substantial toxicities, with several patients reported to have grade 3 or 4 toxicities, most commonly dermatologic (*n* = 14) or metabolic (*n* = 10). One of the patients who had a PR to sorafenib had OCCC and remained progression free for at least 6 months, suggesting that this subtype may be more sensitive to anti-angiogenic treatments [60], although an OCCC selective clinical trial with sorafenib is lacking. Sorafenib treatment can prolong PFS in patients with advanced clear-cell renal-cell carcinoma [83], which shares similar histological features with OCCC, being clear cytoplasm, and some genomic alterations in the SWI-SNF and PI3K pathways [84]. However, treatment with sorafenib is associated with increased toxic effects [83].

Sorafenib used in combination with cytotoxic therapies at best produced modest clinical efficacy but at the expense of substantial toxicities [62,63,64]. A phase II randomised trial to determine the efficacy of 400 mg of oral sorafenib daily in combination with paclitaxel/carboplatin compared to paclitaxel/carboplatin alone in 85 patients with advanced untreated EOC showed no significant difference between responses in PFS [61]. Additionally, patients who received sorafenib in combination with carboplatin/cisplatin were associated with more grade 3 or 4 toxicities, specifically non-haematological skin toxicities, hand-foot syndrome, mucositis, and hypertension [61]. This suggests that sorafenib may present as a therapeutic for patients with OCCC; however, further research is required to reduce the associated toxicities and determine its efficacy in this subtype. 

#### 8.1.2. Sunitinib

Sunitinib, formerly known as SU11248, is a novel multi-targeted ATP-competitive TKI selective for VEGFR and PDGFR. Early phase I trials with sunitinib demonstrated potential for antitumor and antiangiogenic activity in doses delivered above 50 mg daily [85]. In 2006, sunitinib received FDA approval for the treatment of advanced renal cell carcinoma, having also demonstrated therapeutic potential in other tumours such as NSCLC and gastrointestinal stromal tumours [26,86]. In a phase II study of sunitinib in patients with recurrent epithelial cancer (*n* = 30 with 67% serous histology), only one PR was achieved, along with three carbohydrate antigen (CA) 125 responses, whilst the objective response (OR) rate was 3.3%, and 16% of participants with SD [67]. This study further demonstrated the potential benefit of intermittent dosing, correlating with a higher positive outcome.

These findings are further supported by a 2012 phase II clinical trial in the evaluation of dosage delivery of sunitinib in platinum-resistant OC [65]. The OR for the non-continuous and continuous dosage groups were 16.7% and 5.4%, respectively. The therapeutic benefit of 50 mg intermittent sunitinib as a monotherapy in platinum-resistant OC was noted [65].

In a 2018 phase II study, the effects of sunitinib were investigated for the treatment of persistent or recurrent OCCC [66]. With 30 eligible patients, 6.7% (*n* = 2 out of 30) had PR or complete responses (CR), and the study rendered a PFS of 2.7 months and median OS of 12.8 months [66]. All of the mentioned studies suggest modest efficacy and tolerable effects in the treatment of OC. Currently, studies on the effects of sunitinib on OC tumours are impeded due to the absence of validated biomarkers, which are required in order to validate a response to the drug [87]. Further investigation into the antitumour and antiangiogenic effect on OC will be required to establish such biomarkers but the current responses to sunitinib is promising for future studies.

#### 8.1.3. Pazopanib

Pazopanib is another ATP-competitive multi-kinase inhibitor which exerts effects on VEGFR1-3, PDGFR-α and—β, and c-KIT. Previously, pazopanib in combination has seen limited efficacy due to the adverse side effects with no improvement in PFS [68]. In a randomised phase II trial where patients received weekly paclitaxel in combination with either a placebo (*n* = 52) or 800 mg of pazopanib (*n* = 54) every 28 days, the results showed no significant improvements in median PFS (7.5 vs. 6.2 months) and OS (20.7 vs. 23.3 months) for patients receiving paclitaxel and pazopanib compared to paclitaxel alone [68]. Additionally, patient who received combination therapy reported more adverse side effects (20 out of 54, 37%) compared to those who received cytotoxic therapy alone (5 out of 52, 9.6%) [68], and therefore pazopanib is not recommended in combination with cytotoxic therapeutics. 

The efficacy of pazopanib as a maintenance therapy is more promising. In a phase II trial with 36 recurrent OC patients who had elevated levels of CA-125, 800 mg of oral pazopanib daily resulted 36% patients with reduced CA-125 levels by ≥50% and 113 days median duration of response [69]. High grade adverse effects from the trial include alanine transaminase (ALT) elevation (grade 3, 8%), aspartate aminotransferase (AST) elevation (grade 3, 8%), and peripheral oedema (grade 4, 2.77%) [69]. A phase III randomised trial of 940 patients with OC or primary peritoneal cancer (PPC) given either 800 mg of oral pazopanib daily or placebo for 28 days after completing first-line chemotherapy resulted in a slight improvement in the PFS of patients who received pazopanib (17.9 months) compared to the placebo (12.3 months) [70]. However, there was no significant improvement in the OS of patients administered pazopanib in comparison with the placebo [70]. Further research is being completed where pazopanib is used in combination with other maintenance therapies. Current literature on pazopanib is investigating its efficacy with a reduced dose in combination with oral cyclophosphamide (NCT01238770) [88], or fosbretabulin (NCT02055690) [89], to reduce potential adverse side effects.

#### 8.1.4. Cediranib

Cediranib is an oral VEGFR 1–3 and c-KIT inhibitor which showed anti-cancer activity in small cell lung carcinomas [90], prostate cancer [91], glioblastoma [92], and renal cell cancer [93]. In a phase I trials, the maximum tolerated dose of cediranib was 45 mg daily [94]. In a phase 2 trial, 46 patients with recurrent platinum-sensitive (*n* = 16, 35%) or platinum-resistant (*n* = 30, 65%) OC (*n* = 40, 87%), uterine tube cancer (*n* = 1, 2%), or peritoneal cancer (*n* = 5, 11%) received 45 mg of daily cediranib for 28 days [71]. Eleven of the first 15 patients were administered with 45 mg of daily cediranib but due to toxicities had to lower dosing within median 22 days to 30 mg; the remaining patients were started at 30 mg of daily cediranib [71]. In patients with platinum-resistant cancer, six (20%) had a PR, four (13%) had SD, 15 (50%) had progressive disease (PD), and the remaining patients were excluded because of toxicities. In patients with platinum sensitive cancer two (12.5%) had PR, two (12.5%) had SD, six (37.5%) had PD and the remaining six patients were excluded due to toxicities [71]. Disease status was assessed using RECIST and/or CA-125. Grade 3 toxicities observed include hypertension (46%), fatigue (24%), diarrhoea (13%) and grade 4 toxicities were observed in one patient including hypercholesterolemia and central nervous system haemorrhage with the median PFS for both groups being 5.2 months [71].

Current research has begun to focus on the combination of cediranib with olaparib. Olaparib is a potent oral Poly (ADP-ribose) polymerase (PARP) inhibitor, which has previously shown efficacy in BRCA-associated OC, and specifically in HGSC [95]. A Phase II study trialled the efficacy of 400 mg twice daily olaparib alone (*n* = 46) to 30 mg daily cediranib and 200 mg twice daily olaparib (*n* = 44) [72]. PFS for patients receiving olaparib/cediranib improved (17 months) compared to olaparib alone (9 months, hazard ratio: (HR) 0.42; *p* = 0.005). However, more grade 3 and 4 toxicities were observed in patients receiving Olaparib/cediranib compared to the control, most common including fatigue (*n* = 12 vs. 5), diarrhoea (*n* = 10), and hypertension (*n* = 18) [72]. In a follow-up of these patients after the 2016 endpoint, the median PFS remained significantly longer in patients who received cediranib and olaparib (16.5 months) compared to olaparib alone (8.2 months; HR: 0.5; *p* = 0.007) [96]. The OS was not significantly different between olaparib alone compared to olaparib and cediranib. The PFS benefit appeared to be in germline BRCA1/2 mutation (gBRCAm), where there were significant improvements in both PFS (23.7 months vs. 5.6 months, *p* = 0.002) and OS (37.8 months vs. 23.0 months, *p* = 0.047) [96].

The emergence of resistance raises the question about treatment options after progression on a PARPi [73]. The EVOLVE phase II trial investigated olaparib with cediranib in 34 women with recurrent HGSC after a median of five previous lines of treatment. Patients were grouped into one of three groups based on platinum and PARPi statuses: (i) platinum-sensitive after PARPi (*n* = 11), (ii) platinum-resistant after PARPi (*n* = 10); and (iii) progression on chemotherapy after progression on PARPi, namely the exploratory cohort (*n* = 13) [73]. The combination of cediranib and olaparib was well tolerated, however; activity was varied. None of the platinum-sensitive cohort reached an OR, whereas only a few patients in the platinum-resistant (*n* = 2, 20%) and experimental (*n* = 1, 7.7%) cohorts achieved an OR [73]. PARPi progression were accompanied by reversion mutations in *BRCA1*, *BRCA2* or *RAD51B* (19%), *CCNE1* amplification (16%), *ABCB1* upregulation (15%) and *SLFN11* downregulation (7%). Cediranib did not overcome poor treatment response caused by reversion mutations and upregulated *ABCB1*, although the combination with olaparib is tolerable and had some activity in small number of OC following PARPi progression [73]. Hence, cediranib combination studies with larger sample sizes are warranted.

#### 8.1.5. Erlotinib

Aberrant expression of EGFR is common in EOC and is usually associated with a poor prognosis and outcome; however, anti-EGFR therapies have not been successful in improving PFS and OS. Erlotinib, previously OSI-774, is an oral EGFR-specific ATP competitive TKI that has been shown to be potentially effective with EGFR-positive tumours. Thirty-four patients with platinum resistant or refractory EOC received 150 mg of daily erlotinib for up to 48 weeks or until PD or dose-limiting toxicity [74]. PR was observed in two patients lasting 8+ and 17 weeks; 6% had OR (95% CI, 0.7–19.7%), 15 (44%) had SD, and 17 (50%) had PD [74]. The 1-year survival rate and median PFS was 35.3% (95% CI: 19.8–53.5%) and 8 months (95% CI: 5.7–12.7 months), respectively [74]. Although, Erlotinib was well tolerated with limited grade 3 toxicities including diarrhoea, skin and subcutaneous disorders, and an uncommon grade 4 neutropenia detected [74].

In a phase II study, trialled administration of 150 mg erlotinib daily in combination with carboplatin and paclitaxel as a first-line therapy in newly diagnosed patients with advanced OC. Pathological CR was only achieved in 8 (29%) patients following optimal debulking surgery (<1 cm residual disease) and three (13%) after sub-optimal debulking surgery [97]. Overall, the medial PFS was 50.5 months and OS was 53.5 months, additionally no statistically significant correlation was observed between EGFR amplification status and response [97]. This suggests that there is no clinical significance of incorporating erlotinib in first-line therapies.

Hitre et al., administered 150 mg of oral erlotinib with carboplatin and paclitaxel in 30 patients with platinum-sensitive and 14 patients with platinum resistant disease [75]. In the platinum-sensitive group, 10% had CR and 47% had a PR, whereas in the platinum-resistant group 7% (1 of 14) had PR. However, no improvements in PFS or OS were seen with the inclusion of erlotinib [76]. 

A phase III randomised trial compared the PFS of patients with OC, PPC and fallopian tube carcinomas who received 150 mg oral erlotinib daily for up to 2 years (*n* = 420) as a maintenance therapy after first line chemotherapy compared to observation only (*n* = 415) did not show improvements in PFS or OS [77]. 

The EOCTC-GCG 55,041 randomised phase III trial investigated the use of erlotinib as a maintenance therapy and potential predictive biomarkers [77]. The markers tested include somatic mutations in KRAS, BRAF, NRAS, PIK3C, EGFR, and PTEN, together with the expression of EGFR and downstream signalling molecular status (pAKT, pMAPK) and E-Cadherin and Vimentin [77]. FISH analyses were also used to determine EGFR and HER2 gene copy numbers in patients with OC and PPC [77]. The cohort (39.4% of patients) had an overexpression of EGFR; however, this could not be validated as a marker of poor prognosis despite EGFR copy number status (36.7% gene amplification or high levels of copy) having a significant association with worse PFS and OS [77]. Patients who had at least KRAS, NRAS, BRAF, PIK3CA, or EGFR had a longer PFS (33.1 vs. 12.3 months) than patients with wild-type tumours treated with erlotinib [77]. EGFR status did not predict the responsiveness to erlotinib in patients, nor did the other investigated markers [77]. These data suggest that EGFR copy number status may serve more as a prognostic marker than predictive. 

### 8.2. Small Molecule Non-ATP-Competitive Tyrosine Kinase Inhibitors

Type II and Type III inhibitors, known as non-ATP competitive inhibitors, induce a conformational shift in the target enzyme resulting inhibiting kinase function without interacting with the ATP-binding pocket [57,58]. The non-specific nature of Type II and Type III TKIs presents a pharmacologic opportunity in the treatment of various cancers where the use of Type I inhibitors is limited [79]. Here we describe the limited implementation of non-ATP competitive TKIs in the treatment of ovarian cancer.

#### Tivantinib

MET is a protooncogene responsible for the signalling pathway of c-MET, an RTK for hepatocyte growth factor. Although normally expressed in low levels, restricted to cells of epithelial or mesenchymal origin, c-MET is observed to be overexpressed in various cancers. In OC, overexpression of c-MET is significantly associated with poor prognosis and tumour aggressiveness [46]. Formerly known as ARQ-197, tivantinib is an oral non-ATP competitive inhibitor, highly selective for MET [98]. Tivantinib exhibits its mode of action through binding with dephosphorylated c-MET, preventing autophosphorylation [98]. A phase I study in 2017 involving tivantinib and selective mTOR inhibitor Temsirolimus was conducted on patients with advanced solid tumours. In this group, six participants had OC, of which two participants demonstrated PR and SD, and remained within in the study for 10- and 6-months, respectively [78]. A recent study found tivantinib to improve OS in MET-high advanced hepatocellular carcinoma [99]. Further investigation into the action of tivantinib and establishment of positive predictive biomarkers is required.

## 9. Monoclonal Antibodies Targeting Tyrosine Kinases

RTK signalling can also be targeted using specific monoclonal antibodies. These monoclonal antibodies, described below, bind mostly to the extracellular domain of RTK, except for bevacizumab, which acts by selectively binding circulating VEGF. Chimeric monoclonal antibodies contain the syllabus -xi, for example, cetuximab generated from fusion of the variable region of murine antibody and the human constant region. Other monoclonal antibodies have the syllabus -zu to indicate that they are humanised. According to the mechanism of action, RTK monoclonal antibodies may mediate signalling inhibition through ligand blocking, such as seribantumab, which blocks the binding of NRG1 and HER3 [100]. RTK monoclonal antibodies may also inhibit signalling through non-ligand blocking mechanism, for example cetuximab and trastuzumab, bind to EGFR and HER2, respectively, and prevent receptor dimerization, thus leading to receptor function inhibition [101,102,103]. 

### 9.1. Bevacizumab

Bevacizumab (Avastin®) has been approved for OC treatment in the European union by the Committee for Medicinal Products for Human Use on the 23 December 2011 [104]; Australia by the Therapeutic Goods Administration on the 23 February 2012 [105]; and the United States of America by the Federal Drugs Administration on the 14 August 2014 [106]. Bevacizumab is recommended for first-line treatment in patients with advanced EOC in combination with carboplatin and paclitaxel. For recurrent EOC, bevacizumab is also recommended in combination with other cytotoxic therapies like gemcitabine, topotecan or pegylated liposomal doxorubicin depending on previous treatments and status of the patient [105].

Bevacizumab is theorized to work synergistically with standard chemotherapeutics that target angiogenesis [107,108]. In the tumour microenvironment, carboplatin induces the expression of VEGF, potentiating the anti-angiogenic effects of paclitaxel and bevacizumab through inhibition of endothelial cell migration and proliferation [109]. In 2011, two phase III randomised trials, ICON-7, and GOG-0218, published a significant improvement in PFS when bevacizumab was used in-combination with standard first-line therapies.

ICON-7 was an international, randomised, open-label phase III trial of 1528 women with optimally debulked stage III/IV or high-risk early-stage EOC [110]. In the high-risk population, treatment with bevacizumab was associated with a significant improvement in PFS, 16 months, compared to the 10.5 months for the control group (HR: 0.73, 95% CI: 0.61–0.88; *p* < 0.001) [110]. Additionally, treatment with bevacizumab was significantly associated with improvements in OS, 39.5 months, when compared to controls, 30.2 months (HR: 0.78; 95% CI: 0.63–0.97) [110].

GOG-0218 was a multicentre, randomised placebo-controlled Phase III trial of 1873 patients with advanced EOC [111]. This trial had two phases: the initial cytotoxic therapy where all patients received carboplatin/paclitaxel in combination with placebo or bevacizumab, followed by a maintenance phase. Patients were randomised into one of three groups: placebo received throughout carboplatin/paclitaxel phase and maintenance phase (control cohort); bevacizumab received in combination with carboplatin/bevacizumab and transferred to placebo for maintenance (initial cohort); and bevacizumab received throughout carboplatin/paclitaxel and maintenance phases (continuous cohort) [111]. A 3.8 month improvement of PFS was seen when comparing patients in control (10.3 months) to the continuous cohort (14.1 months; HR: 0.72; 9% CI: 0.63–0.82) [111]. Patient who received bevacizumab in combination with carboplatin and paclitaxel saw a one month increase in PFS when compared to control, however this was not statically significant (11.2 vs. 10.3 months; HR: 0.91; 95% CI: 0.80–1.04). Median follow up of patients was 103 months after which approximately 80% of patients had died. Comparison of all three groups saw no difference in OS, which was approximately 41–43 months. Additionally, patients with stage IV EOC receiving bevacizumab throughout had a significant improvement in OS (42.8 months) when compared to the stage IV controls (32.6 months), and stage IV who received initial bevacizumab (34.5 months) [111]. In both trials, bevacizumab did not affect quality of life with no difference overall in the ICON-7 trial; however, in GOG-0128 initially quality of life was slower in patients receiving bevacizumab, but no difference was detected during maintenance therapy [110,111].

The AURELIA phase III was a randomised trial of three different chemotherapeutics, paclitaxel, pegylated liposomal doxorubicin, and topotecan, in combination with or without bevacizumab [112]. The addition of bevacizumab to each chemotherapeutic resulted in an increase in PFS, for topotecan cohort median PFS was 5.4 vs. 2.1 months (HR: 0.32; 95% CI: 0.21–0.49), paclitaxel cohort median PFS was 10.4 vs. 3.9 months (HR: 0.46; 95% CI: 0.30–0.71), and pegylated liposomal doxorubicin PFS was 5.4 vs. 3.5 months (HR: 0.57; 95% CI: 0.39–0.83) [112]. However, interpretation of these findings needs to be handled with caution as for each cohort the chemotherapy group was not randomised [112].

It is thought that antiangiogenic agents may increase the efficacy of PARP inhibitors through reducing the homologous recombination repair genes and proteins [113]. Hence, there are several ongoing trials focusing on the efficacy of bevacizumab in combination with PARP inhibitors. PALO-1, a phase III randomised control trial of newly diagnosed EOC patients who are undergoing bevacizumab maintenance therapy in combination with olaparib showed an increase in PFS was seen in patients receiving both olaparib and bevacizumab compared to bevacizumab alone, 22.1 months vs. 16.6 months, respectively (HR: 0.59; 95% CI 0.49–0.72; *p* < 0.0001) [114]. Additionally, the combined treatment was well tolerated with toxicities between the two groups remaining relatively consistent [114].

ENGOT-OV24/ANAOVA (NCT02354131) is a phase II trial assessing the efficacy of niraparib in combination with bevacizumab as compared to niraparib alone in platinum-sensitive EOC patients receiving maintenance therapies [115]. Bevacizumab in combination with niraparib showed improvements in PFS (11.9 months) when compared to niraparib only maintenance therapies (5.5 months; HR: 0.35; *p* < 0.001) [115]. This data may provide a new combined molecular therapeutic option for newly diagnosed patients who cannot receive chemotherapy [116,117]. Currently, trials of bevacizumab in combination with rucaparib, NCT03462212 [118], and niraparib, NCT03326193 [119], in patients with newly diagnosed EOC are ongoing.

### 9.2. Cetuximab

Cetuximab, C225, is a chimerised monoclonal antibody which targets EGFR through binding to the extracellular domain of the receptor, preventing ligand activation [120]. A phase II trial of weekly cetuximab (initially 400 mg/m^2^, followed by weekly 250 mg/m^2^) in two 3-week cycles was administered in 25 patients with EOC, PPC or UTC, one patient had PR (4%), and nine had SD (36%) with a median PFS of 2.1 months and a 1-year survival rate of 54.8%. Additionally, serum testing was completed prior to and at the end of the trial to determine potential protein markers for drug activity, which showed 12 proteins (CA 72-4, cytokeratin 19, Fibrinogen, Growth Hormone, Human Epididymis Protein 4 (HE4), Heat Shock Protein 27, Interleukin-6, Interleukin-8, Kallikrein-related Peptidase 10, Matrix Metalloproteinase-7, Serum Amyloid A and Tumour Necrosis Factor α) were elevated in the PD group relative to the PR/SD group [121].

Another phase II trial of cetuximab (initially 400 mg/m^2^, followed by weekly 250 mg/m^2^) was undertaken but in combination with carboplatin in 29 patients with recurrent platinum-sensitive OC or PPC (26 had EGFR positive tumours) resulted in three patients with CR (10.34%), six with PR (20.69%) and eight with SD (27.59%) [122]. The median PFS was 9.4 months, although cetuximab had a moderate response rate, this study did not meet set criteria to move onto further testing. Research has shifted away from cetuximab in OC treatment as no phase III trials are being currently undertaken.

### 9.3. Pertuzumab

Pertuzumab is a recombinant humanised monoclonal antibody which inhibits ligand-activated heterodimerisation of HER2 with other ErbB receptors [123]. Direct inhibition of HER2 association with partner receptors results in the blocking of signalling cascades, making the cells more susceptible to chemotherapy-induced apoptosis [124].

A 2006 phase II study tested the effects of pertuzumab as a single agent in tumours with progression post-platinum-based therapy [125]. The median PFS was longer in HER2-positive tumours than non-HER2 positive, being 20.9 and 5.8 months, respectively. Although not statistically significant, this clinical activity demonstrated therapeutic potential for pertuzumab use in patients with heavily pre-treated OC and suggested the importance to consider HER2 status [125] 

A phase II study demonstrated the effects of pertuzumab in combination with gemcitabine in platinum-resistant OC [126]. A trend of improved PFS with participants treated with pertuzumab was established, and a higher objective response rate for the experimental group compared to the control group, 13.8% and 4.6%, respectively. The study established potential predictive biomarkers for pertuzumab in OC, with findings suggestive of pertuzumab being active and tolerable in platinum-resistant OC tumours with low HER3 mRNA expression [126]

Following clinical trial success in breast cancer treatment, a 2016 phase III study assessed the therapeutic potential of pertuzumab combined with paclitaxel [127]. PFS benefit was observed in patients with platinum-resistance (PFI: 3–6 months, *p* = 0.02) compared to platinum-refractory patients. The effects of pertuzumab on PFS were further pronounced in patients with no prior antiangiogenic therapy exposure. Interestingly, no therapeutic benefit was associated with HER3 mRNA expression levels, in contrast to previous reports [125,126,127]. Clinical trials of pertuzumab in OC treatment have proven promising, but future research would be enhanced with the establishment of biomarkers predictive of therapeutic efficacy.

### 9.4. Trastuzumab

Trastuzumab is a high affinity humanised monoclonal anti-HER2 antibody. Specific to the extracellular domain of HER2, the drug is most effective in tumours with an overexpression of HER2, occurring in only 11% of OC cases [128]. Approved for use in refractory metastatic HER2-positive breast cancer, the drug is used as both a single agent and in combination with paclitaxel and has demonstrated significant improvement to disease-free survival [25]. The effects of trastuzumab on OC have been investigated through a number of clinical trials with varying success. A phase II trial conducted in 2003 administered weekly doses of trastuzumab for up to 53 weeks to recurrent or refractory OC, guided by HER2 positive status. The complete and partial response rates were 2% and 5%, respectively [129]. A 2008 phase II trial combining trastuzumab with paclitaxel and carboplatin in the treatment of advanced HER2-positive OC showed a CR rate of 43% and a SD of 29%. This study only tested 7 participants, thus decreasing the reliability of the findings [130]. A 2009 study showed that HER2 amplification is relatively common in ovarian mucinous carcinomas (18.2%) [39]. This small study consisted of trastuzumab treatment in three patients with recurrent mucinous carcinomas which had HER2 overexpression and amplification. One patient showed significant response, with an additional participant exhibiting a partial response. Trastuzumab was administered both with and without paclitaxel and the results indicate promising efficacy in this usually non-responsive OC subtype.

A common theme from trastuzumab use in OC is the lack of HER2 overexpression limiting the therapeutics success, unlike in breast cancer. This may be due to the drug mechanism varying from its previously studied action in OC and warrants further investigation. Minimal studies on trastuzumab as a single agent therapy in OC have been conducted, but the findings of these earlier clinical trials indicate a degree of therapeutic potential that may be beneficial if investigated further.

### 9.5. Seribantumab

Seribantumab, formerly MM-121, is a fully human immunoglobulin G2 mAb, tasked with targeting HER3 [131]. It elicits its mode of action through competing with Neuregulin 1 (NRG1) in binding within the extracellular domain of HER3, antagonising the receptor signalling. NRG1 binding drives HER3 heterodimerization with other ErbB family members, most notably HER2, activates the PI3K/Akt growth and survival signalling pathways, and mediates increased insensitivity to applied therapeutics [131]. *NGR1* gene fusion is a potential oncogenic driver; however, the occurrences in solid tumours are rare, with only 0.2% in all solid tumours and 0.4% found in OC [132].

In a 2016 randomised phase II study, seribantumab was experimentally combined with paclitaxel and administered to 140 participants, compared to a controlled standard dosing of paclitaxel alone [133]. The median OS for the experimental group was 13.78 months, compared to 10.12 months for the control, rendering these findings insignificant [133]. The PFS did not prove statistically significant between the experimental and control groups, being 3.75 and 3.68 months, respectively [133]. Whilst the OR was not met for the wider population, a potential biomarker was observed in the patients with tumours positive for the NGR1 alteration, and low levels of HER2. In these biomarker-positive participants, the combination of seribantumab and paclitaxel rendered a more positive outcome, with a PFS HR of 0.37. This was reversed in the biomarker-negative subgroup who favoured paclitaxel alone. The findings of this study demonstrated the use of NGR1/low HER2 as predictive biomarkers for the strategic use of seribantumab in further studies [133]. These findings were further validated preclinically with the successful regression of patient derived NGR1-positive HGSOC tumours in mice [134]. Administered with a range of seribantumab doses twice weekly over 27 days, 100% of the mice model demonstrated significant tumour shrinkage, with only one case of tumour regrowth was observed after treatment from the highest dose subgroups, 5 mg and 10 mg [134]. The clinical potential of seribantumab in the treatment of OC and other solid tumours with NGR1 alterations is currently under further investigation in a phase II clinical trial, NCT04383210 [86].

## 10. Immunotherapeutic Targeting Tyrosine Kinases

Despite the efficacy across various cancers, passive immunity poses limitations to the use of mAbs. In response to this, an emergence of vaccines has occurred, aimed at inducing active immunity against tyrosine kinase receptors. The targets of these vaccines vary widely, including specific tumour antigens, protein components of the receptor extracellular domain, epitope chimeras, and DNA sequences [135]. Current vaccines under clinical investigations have been used to target trastuzumab- and pertuzumab-like binding sites. The current approaches include both the use of the vaccine to enhance current therapies or as a single agent. In a current phase I trial (NCT01376505), peptide B-cell epitope vaccines are being implemented to represent the effects of trastuzumab and pertuzumab on their targeted binding sites. In a 2019 report on the study, the B-cell vaccine aimed to stimulate the patients’ immune system to elicit a polyclonal antibody response against the targeted binding sites, with the patients’ antibodies inhibiting HER-2 receptor phosphorylation [136]. The findings thus far from the study indicate increased cell death, like that of trastuzumab, a promising sign in favour of anti-tumour vaccines [136]. Another current phase I clinical trial (NCT00436254) is investigating the side effects and best dose of vaccine therapy in combination with sargramostim, an immunostimulatory based on granulocyte macrophage colony-stimulating factor, in stage III-IV HER-2 positive breast cancer and OC [137]. Similar to the single agent, this plasmid-based DNA vaccine aims at eliciting specific HER2 immune responses [137]. The vaccine will by-pass the need for repeated treatments that are currently required of oral therapeutics, as well as provide therapeutic intervention for a lower cost with lower toxicities. Further studies are required to validate the efficacy of the vaccine approach in OC, but phase I clinical trials have so far demonstrated promising therapeutic potential. 

## 11. Future Prospective/Conclusions

Most of the targets for the small molecule inhibitors include VEGFR1-3, PDGFR, EGFR, c-KIT, c-RAF, and MET [57]. Despite numerous trials, the overall outcome of small molecule inhibitors alone and in combination with chemotherapy have been disappointing, summarised in Table 1. The majority of small molecule TKI drugs are well-tolerated; however, their efficacy in improving the PFS and OS in patients with platinum-sensitive and platinum-resistant EOC has had limited success. Three trials were able to produce statistically significant results in respect to their trial aims: NCT00073307 sorafenib as a monotherapy [83], NCT00866697 pazopanib as a monotherapy following first-line chemotherapy [70] and NCT01116648 cediranib in combination with olaparib [72].

Monoclonal antibodies, summarised in Table 2, have been shown to be a more promising class of drugs to treat EOC, with bevacizumab being recommended to most women with recurrent EOC [104,105,106]. Bevacizumab can be prescribed both in combination with cytotoxic therapies and as a maintenance therapy to improve PFS; however, statistically significant improvements in PFS were seen whilst used in conjunction with cytotoxic agents [112,114,115]. Other monoclonal antibodies have not had as much success as bevacizumab, with the majority not progressing to phase III trials.

A limiting factor for the targeting of RTKs in OC is the lack of an established OC-specific model. Many current and previously conducted therapeutic studies on OC are based on the responses of other cancers such as NSCLC, breast cancer, prostate cancer, and thyroid cancer. The genomic landscape of EOC differs from these cancers and is complicated by different histotypes harbouring separate mutational profiles. Furthermore, EOC can be divided into several histological and molecular subtypes [2]. Therefore, there is increasing complexity in treating EOC due it’s heterogeneous nature. Complete profiling of tyrosine kinase targets in the different histological/molecular subtypes is warranted and may provide insight into new potential targets or therapeutic combinations. Other emerging RTK targets such as Axl family and ROR family are being investigated preclinically and may be further developed. 

Previously, mAbs targeting HER2/HER3, such as pertuzumab [118,125,126,127], trastuzumab [129,130], and seribantumab [133], showed some activity; however, their use is limited by the low frequency of HER2 overexpression and a lack of predictive biomarkers to select tumours that would be responsive. There are several clinically used biomarkers in OC that can assist in diagnostic or disease monitoring, however, ones which may predict the outcome or response of a patient to certain therapeutics, especially to therapeutics targeting the kinases, have remained relatively underdeveloped. Currently, heightened CA-125 concentration is the main biomarker used to monitor disease progression, however, it has low sensitivity in detecting early stage OC as well as elevated levels seen in several other pathological and non-pathological conditions such as endometriosis, pregnancy, pelvic inflammatory disease and menstruation [138,139]. Additionally, HE4 has a higher affinity to detect serous OC compared to other histological subtypes and provides increased specificity in OC detection and management as serous OC is the most widely diagnosed [140,141]. When used in combination, HE4 and CA-125 have a combined sensitivity of 92.9% at a 95% specificity compared to 78.6% sensitivity for both HE4 and CA-125 alone [141]. Development of Risk of Malignancy (RMI) and Risk of Ovarian Malignancy Algorithms (ROMA) were developed to amend the innate characteristics of these markers.

There is emerging, though still limited, research discussing predictive biomarkers regarding patient response to specific tyrosine kinase therapeutic drugs, which should improve strategy to treat OC by better capturing the heterogeneity of this disease. 

The new focus of therapeutic targets for EOC are combining small molecule or monoclonal antibodies with drugs that have more established efficacy i.e., PARP inhibitors. For example, phase II-III clinical trials, ENGOT-OV24/ANAOVA [115] and PALO-1 [114], examined combination of bevacizumab and PARP inhibitor, niraparib and olaparib, respectively, in platinum-sensitive tumours. Both trials reported significant benefit for the combination with improved PFS in combination as compared to single agent alone. This combination was also well tolerated [114,115], and thus shows promise for platinum-sensitive OC patients.

Other noted combination includes cediranib and olaparib [72], where there was PFS improvement in women with recurrent platinum-sensitive high-grade serous or endometrioid OC compared to olaparib monotherapy, although further studies are required to better understand the quality of life and patient-reported outcomes of this combination regimen as grade 3 and 4 adverse events were more common in the combination group [72,96]. 

As passive immunity poses limitations to the use of monoclonal antibodies, the emergence of vaccines as both a single agent and in combination to enhance current therapies is being trialled. The peptide B-cell vaccine has shown promise in phase I trials in patients with advanced solid tumours [136], with further trials investigating this avenue of therapeutics currently occurring [137], it may expand the arsenals of therapy to improve treatment of recurrent OC.

In summary, although numerous trials have examined tyrosine kinase inhibitors in OCs, further studies are required for more effective drugs to improve patient outcomes. Several limitations include the difficulty of targeting tyrosine kinases with low frequency alterations in OC and a lack of predictive biomarkers to stratify patients. Noting histological subtypes to match treatment is an important consideration moving forward. There is still a limited option for platinum-resistant OC. Future work on emerging tyrosine kinase targets, effective combination approaches, and treatment involving active immunity may improve current therapy.

## Figures and Tables

**Figure 1 biomedicines-10-02113-f001:**
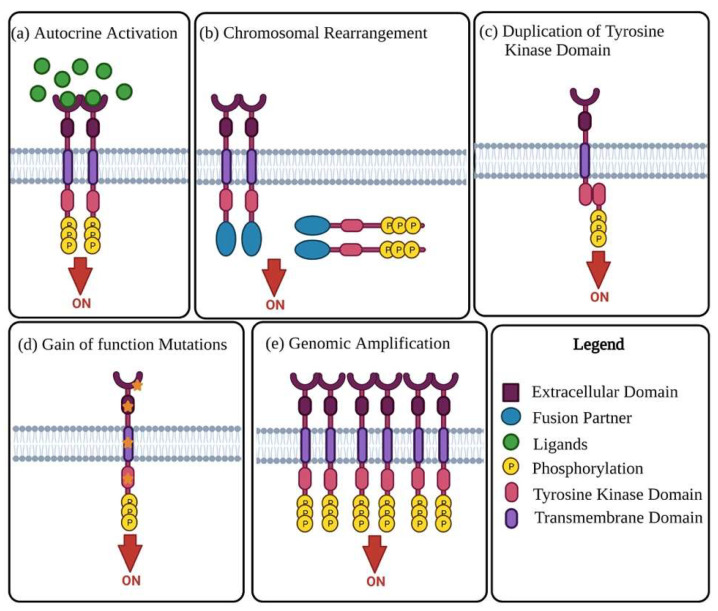
Mechanisms of oncogenic RTK activation. (**a**) Visual representation of **autocrine activated RTK signalling**. Increased ligand production by cancer cells or the tumour microenvironment causes activation of RTK signalling, leading to increased kinase activity and phosphorylation of the C-terminal tail of the receptor. (**b**) **Chromosomal rearrangement** results in the creation of a hybrid fusion oncoprotein composing partly of the TK and fusion partner. These RTK fusion partners are often cytoplasmic or membrane bound proteins depending on the position of the genomic cut-off point. The rearrangement results in deletion of regulatory domains, which then causes tyrosine kinase activation. (**c**) **Duplication of tyrosine kinase domain** could result in the formation of an intramolecular dimer and activation of RTK, in the absence of ligands. (**d**) Diagrammatic illustration of probable **gain-of-function mutations** in several RTK subdomains. Mutations in these regions lead to constitutive activation of the RTK, more often than not in the absence of ligands. (**e**) Visual representation of RTK **genomic amplification** is frequently a consequence of the genomic amplification of RTK genes, resulting in the increase of local concentrations of RTKs.

**Table 1 biomedicines-10-02113-t001:** Clinical trials involving small molecule tyrosine kinase inhibitors in ovarian cancer.

Drug	Trial ID	Clinical Trial Phase	Cohort Size	Dose	Side Effects	Outcome	References
Sorafenib	NCT00093626	Phase II	*n* = 71	400 mg twice daily in 4-week cycles until PD or intolerable toxicity	Grades 1–3 dermatologic (76%) and gastrointestinal (79%) AEs most common.	3.4% PR and 33.9% SD after 6 months	[60]
		Phase II	*n* = 85	Paclitaxel and carboplatin every 21 days for a maximum of 6 cycles, with or without 400 mg sorafenib twice daily.	Increased toxicities with sorafenib. Grade 3/4 non-haematological skin toxicities, hand-foot syndrome, mucositis, and hypertension.	No significant difference in OR, PFS or OS	[61]
	2006-004644-24	Phase II	*n* = 4	400 mg twice daily alongside carboplatin/paclitaxel schedule.	Fatigue, anorexia, diarrhoea, rash, and hand-foot skin reaction.	Study terminated on the recommendation of independent safety monitoring board.	[62]
	NCT00526799	Phase I/II	Phase I: *n* = 16. Phase II: *n* = 14	Phase I: 400 mg and 800 mg daily with weekly reducing topotecan dosage. Phase II: 400 mg sorafenib daily with topotecan 3.5 mg/m^2^ weekly.	Grade 3/4 AEs: leukopenia/neutropenia, thrombocytopenia, anaemia, fatigue, nausea, and vomiting	Phase I: 4 PR. Phase II: 1 PR. OR: 16.7%. 46.7% SD with median OS 14.0 mths	[63]
		Phase II	*n* = 43	400 mg twice daily with cyclic dosage of gemcitabine	Hand-foot syndrome, fatigue, diarrhoea, and hypokalaemia.	4.7% PR; 2.3% SD with median PFS and OS 5.4- and 13.0 mths, respectively.	[64]
Sunitinib	2007-003089-16	Phase II	*n* = 73	Either 50 mg once daily for 4 weeks in 6-week cycles or 37.5 mg once daily continuously	Grade 3/4 AEs: 46 in continuous and 60 in non-continuous groups. Includes: haematologic aberrations, and gastro-intestinal syndrome.	OR was 5.4% and 16.7% in continuous and non-continuous dosage groups, respectively. No significant difference in PFS and OS.	[65]
	NCT00979992	Phase II	*n* = 30	50 mg once daily for 4 weeks in 6-week cycles.	Grade 3 AEs: fatigue, haematologic aberrations, abdominal pain and hypertension. Grade 4–5 AEs: acute kidney injury, allergic reactions, haematologic aberrations and stroke,	6.7% had PR or CR. Median PFS and OS was 2.7 and 12.8 mths, respectively.	[66]
	NCT00979992	Phase II	*n* = 30	50 mg once daily for 4 weeks in 6-week cycles.	Grade 3 AEs of fatigue, mucositis, nausea, hand-foot syndrome, diarrhoea, and hypertension. No grade 4 AEs.	Median overall PFS of 4.1 mths 16 with SD and 1 PR	[67]
Pazopanib	NCT01468909	Phase II	*n* = 106	800 mg every 28-days, with weekly paclitaxel chemotherapy	Grade 3/4 AEs include: neutropenia and hypertension.	No significant change in median PFS or OS. 66% PD in placebo patients compared to 31.5% receiving pazopanib.	[68]
	NCT0281632	Phase II	*n* = 36	800 mg once daily	Grade 3 ALT & AST elevation in 8%. Grade 4 peripheral oedema in 2.8%	Decreased CA-125 in 31% patients. Median response time and duration was 29- and 113-days, respectively. OR: 18%	[69]
	**NCT00866697**	**Phase III**	** *n* ** ** = 940**	**800 mg** **once daily, 28** **-** **days following first-line chemotherapy**	**Grade 3** **/** **4 AEs of hypertension (30.8%), neutropenia (9.9%), liver-related toxicity (9.4** **%),** **and diarrhoea (8.2%).**	**PFS for pazopanib and placebo was 17.9 and 12.3 mths respectively (*p* = 0.0021). No signifi-cant difference in OS.**	[70]
Cediranib	NCT00275028	Phase II	*n* = 46	45 mg daily, lowered to 30 mg daily	AEs (all grades) include: diarrhoea, voice changes, hypertension, fatigue, headaches, hypothyroidism, mucositis, and nausea	PR was observed in 17% of patients, with a further 13% with SD. 45% had disease progression.	[71]
	**NCT01116648**	**Phase II**	** *n* ** ** = 44**	**Olaparib** **200 mg** **and** **Cediranib 30 mg** **twice daily**	**Grade 3 AEs: fatigue, diarrhoea and hypertension. Grade 4 AEs: hypertensive crisis and myelodysplastic syndrome**	**Median PFS** **was 16.5 mths** **for combination therapy vs** **.** **8.** **2 mths in control** **(*p* = 0.007). OS at** **24 mths** **for combination vs** **.** **control was 81% and 65% respectively.**	[72]
	NCT02681237	Phase II	*n* = 34	200 mg Olaparib combined with 20 mg Cediranib twice daily	Most common all-grade AEs include diarrhoea, nausea, vomiting, and fatigue, mainly all grade 1 or 2.	2 PR and 4 SD (6 total) in platinum resistant patients, 9 SD but no OR in platinum sensitive patients. 1-year OS for platinum sensitive and resistant was 82% and 69%, respectively.	[73]
Erlotinib		Phase II	*n* = 28	150 mg daily with carboplatin and paclitaxel	Grade 3 diarrhoea, skin and subcutaneous disorders 1 case of grade 4 neutropenia	42.9% SD and 5.7% PR with PFS and 1-yr survival rate of 8 mths and 35.5%, respectively. 28.6% CR in optimally debulked patients with median PFS and OS of 80.5 and 53.5 mths, respectively. However not clinically significant	[74]
	NCT00030446	Phase II	*n* = 50	150 mg orally with carboplatin and paclitaxel	Grade 3/4 dry skin (1.7%), abdominal pain (2.4%) and increase of gamma-glutamyl transpeptidase (3.4%)	Platinum sensitive patients had 10% CR and 47% PR Platinum resistant patients had 7% PR only No improvement in PFS or OS.	[75]
	NCT00263822	Phase III	*n* = 835	150 mg daily (orally) as maintenance following chemotherapy	Diarrhoea, loss of appetite, nausea/vomiting, and fatigue.	No improvement in PFS or OS	[76,77]
Tivantinib	NCT01625156	Phase I	*n* = 6	120 mg Tivantinib twice daily with weekly 20 mg IV temsirolimus, increasing exponentially after 28-day cycles.	Grade 2 AEs include: anaemia, fatigue, anorexia, and hypoalbuminemia. Grade 3 AEs: anaemia, hypophosphatemia, hypertension, and hyponatremia. Grade 4 neutropenia in 2 patients	1 PR 1 SD	[78]

Note: Trials in bold produced statistically significant results in respect to the trial aims.

**Table 2 biomedicines-10-02113-t002:** Clinical trials involving monoclonal targeting tyrosine kinase pathways in ovarian cancer.

Drug	Trial ID	Clinical Trial Phase	Cohort Size	Dose	Side Effects	Outcome	References
Bevacizumab	**2005-003929-22**	**Phase III**	** *n* ** ** = 1528**	**7.5 mg/kg of body weight in combination with carboplatin/paclitaxel chemotherapy, every 3 weeks for 12 cycles.**	**Adverse effects include grade 1–2 muco-cutaneous bleeding (36%), grade 2+ hypertension (18%), grade 3 thrombo-embolic events (7%), and gastrointestinal perforations (1%).**	**Significant improvement in PFS with 16 mths, compared to control group 10.5 mths (*p* < 0.001). Improved OS, 39.5 mths**	[110]
	NCT0026287	Phase III	*n* = 1873	6x 21-day cycles of carboplatin/paclitaxel chemotherapy with the addition of 15 mg/kg body weight bevacizumab or placebo	Most common AEs include pain, hypertension, and neutropenia, all ≥ grade	PFS improved by 3.8 mths with continuous cohort. No significant difference in median OS. Increase OS in grade IV EOC receiving continuous cohort (42.8 mths) vs. grade IV EOC control (32.6 mths)	[111]
	**NCT00976911**	**Phase III**	** *n* ** ** = 361**	**Paclitaxel, PLD and** **topotecan** **treatments +/- bevacizumab** **10 mg** **/kg body weight every 2 weeks**	**Increased incidence of hypertension and proteinuria ≥grade 2 was observed with bevacizumab, along with ≥grade 2 GI perforation and fistulas**	**Median PFS increased to 6.7 mths for continuous (*p* < 0.001). Median OS increased to 16.6 mths (*p* < 0.174). OR of 27.3% with continuous (*p* = 0.001)**	[112]
	**NCT02477644**	**Phase III**	** *n* ** ** = 806**	**Olaparib (300 mg twice daily) with and without bevacizumab (15 mg/kg body weight every 3 weeks)**	**The most common AEs experienced in the experimental group were fatigue, nausea, and anaemia (all grades)**	**Increased PFS with combination, 22.1 mths vs. 16.6 mths without (*p* < 0.001)**	[114]
	**NCT02354131**	**Phase II**	** *n* ** ** = 97**	**Niraparib (300 mg) once daily, with bevacizumab (15 mg/kg) once every 3 weeks until disease progression**	**Combination therapy associated with AEs proteinuria (21%) and hypertension (56%), both of any grade)**	**Increase PFS 11.9 mths vs. 5.5 mths with and without bevacizumab respectively (*p* < 0.001)**	[115]
Cetuximab	NCT00082212	Phase II	*n* = 25	21-day cycles of 400 mg/m^2^ initial dose, followed by weekly 250 mg/m^2^ doses of cetuximab.	Grade 3 AEs of arthralgia, headache and acneiform rash were recorded. Chills, nausea, stomatitis, diarrhoea, and constipation also were noted, at all grades.	4% PR with another 36% SD. PFS 2.1 with a 1-year survival rate of 54.8%	[121]
	NCT00086892	Phase II	*n* = 29	Cyclic 400 mg/m^2^ and 250 mg/m^2^, combined with carboplatin	Most common AEs grade 3 dermatologic toxicity (32%), thrombocytopenia (14%) and metabolic toxicity (14%).	CR and PR was observed in 11.5% and 23% of patients respectively. A further 30.8% had SD. 11.5% had PD, and 30.8% not evaluated.	[122]
Pertuzumab		Phase II	*n* = 123	Cohort 1- 840 mg loading dose, then 420 mg on day 1 of each 3-week cycle. Cohort 2- 1050 mg on day 1 of each 3-week cycle	Most common AE was diarrhoea, mainly grade 1–2. Other AEs include fatigue, rash/dermatitis, nausea, and abdominal pain across both cohorts.	PR in 3.6% and 4.8% for cohort 1 and 2 respectively. 4 patients in each cohort had SD. Median PFS of 6.6 weeks. Median OS 52.7 weeks	[125]
	NCT00096993	Phase II	*n* = 130	Gemcitabine 800 mg/m^2^ on days 1 and 8 of 21-day cycle with Pertuzumab, loading dose 840 mg followed by 420 mg every 3 weeks.	Higher incidence of AEs in pertuzumab group, including fatigue, nausea, diarrhoea, and backpain. Increased grade 3–4 neutropenia, thrombocytopenia, back pain, and diarrhoea in pertuzumab group.	Median PFS higher in combination group than control (2.9 mths vs. 2.6 mths, *p* = 0.07). 13.8% PR observed with combination therapy. Median OS similar between groups (13.1 mths placebo vs. 13.0 mths combination, *p* = 0.65) Increased benefit of combination therapy with HER3 expression (*p* = 0.0002)	[126,127]
	NCT01684878	Phase III	*n* = 156	Chemotherapy delivered (topotecan, paclitaxel or gemcitabine) with pertuzumab 840 mg loading dose followed by 420 mg every 3 weeks.	Common AEs for combination therapy were fatigue, nausea, diarrhoea, neutropenia, and anaemia.	Median PFS 4.3 mths (*p* = 0.14). PFS benefit for platinum resistant patients as but not in platinum-refractory (*p* = 0.02)	[118,128]
Trastuzumab		Phase II	*n* = 41	Initial dose of 4 mg/kg, then weekly at 2 mg/kg intravenously	Mild toxicities, tolerated well by patients with no treatment related deaths.	OR of 7.3%. 1 CR and 2 PRs recorded. PFS was 2.0 mths	[129]
		Phase II	*n* = 320	Paclitaxel and carboplatin with trastuzumab in 8 mg/kg initial dose, followed by 6 mg/kg for subsequent cycles, every 3 weeks.	Most common AEs were febrile neutropenia, grade 3 infection, and grade 2 neurotoxicity	3 patients had CR, with a further 3 observed to have SD. Median PFS and OS were 2.9- (range: 1.5–44.2) and 12.3 mths (range:1.9–44.2), respectively.	[130]
Seribantumab	NCT01447706	Phase II	*n* = 223	Paclitaxel combined with either Seribantumab (40 mg/kg initial load, followed by 20 mg/kg weekly) or placebo	Increase in AEs observed in combination therapy, including diarrhoea, fatigue, nausea abdominal pain, hypokalaemia, and anaemia.	Median OS 13.75 compared to 10.12 for control group (*p* = 0.972). Median PFS with seribantumab 3.75 (*p* = 0.864). Identified potential biomarkers	[133]

Note: Trials in bold produced statistically significant results in respect to the trial aims.

## Data Availability

Not applicable.

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
