# Peer review of "Targeting Tyrosine Kinases in Ovarian Cancer: Small Molecule Inhibitor and Monoclonal Antibody, Where Are We Now?"

_biomedicines, 2022, doi:10.3390/biomedicines10092113_

Round 1

Reviewer 1 Report

Dear authors

The present review discusses an important topic related to ovarian cancer. The review is well written, however, some comments are to be considered when revising the manuscript.

1. Ovarian Cancer: Current Understanding and Treatment

In this paert, authors reported statistics related to Australia. It'd be better to report general (world) statistics.

3. ERBB Receptors

"A common family of RTKs ......... differentiation and survival."

A reference should be added.

8. Small Molecule Tyrosine Kinase Inhibitors 

A figure summarizing how different Tyrosine kinase inhibitors may target different signalling pathways could be of great interest to the readers. 

Table 1. Clinical trials involving small molecule tyrosine kinase inhibitors in ovarian cancer.

Although the information given is interesting, I think that the type of the study, population and outcome should be added. 

8.1. Small Molecule ATP-competitive Tyrosine Kinase Inhibitors

An introductury paragraph as a general overview of these molecules should be added. 

8.2. Small Molecule Non-ATP-competitive Tyrosine Kinase Inhibitors 

An introductury paragraph as a general overview of these molecules should be added. 

9. Monoclonal antibodies targeting tyrosine kinases

An introductury paragraph as a general overview of these molecules should be added. 

Table 2. Clinical trials involving mAb targeting tyrosine kinase pathways in ovarian cancer

Although the information given is interesting, I think that the type of the study, population and outcome should be added. 

Reviewer 2 Report

Please see attached peer review report

Round 2

Reviewer 2 Report

I am satisfied with the edits performed by the authors